# Arzanol: A Review of Chemical Properties and Biological Activities

**DOI:** 10.3390/plants14223474

**Published:** 2025-11-14

**Authors:** Yulian Voynikov

**Affiliations:** Department of Chemistry, Faculty of Pharmacy, Medical University, 1000 Sofia, Bulgaria; y_voynikov@pharmfac.mu-sofia.bg

**Keywords:** arzanol, α-pyrone, phloroglucinol, anti-inflammatory, antioxidant, SIRT1, neurobehavioural, autophagy modulation

## Abstract

Arzanol, a prenylated phloroglucinol–α-pyrone heterodimer, displays a broad range of pharmacological properties. This review compiles findings from 2007 to 2025 on its chemistry, conformational behavior, bioactivities, molecular targets, and pharmacokinetics. Arzanol shows potent anti-inflammatory activity through NF-κB inhibition and dual suppression of mPGES-1 and 5-LOX, antioxidant and cytoprotective effects via radical scavenging and metal chelation, and selective antibacterial activity. Arzanol also modulates autophagy, mitochondrial function, and metabolic pathways, with high-affinity binding to brain glycogen phosphorylase and SIRT1. Pharmacokinetic data indicate gastrointestinal stability, intestinal absorption, and limited blood–brain barrier penetration. *In vivo*, arzanol exhibits neuroprotective, neurobehavioral, and metabolic effects, while showing selective cytotoxicity toward cancer cells with minimal impact on normal cells. This review evaluates the diverse biological activities of arzanol, analyzing the relationship between its unique conformational flexibility and multitarget pharmacological effects.

## 1. Introduction

Arzanol (Figure 1), a prenylated phloroglucinol α-pyrone heterodimer, has emerged as a multifunctional bioactive natural product with diverse pharmacological properties. First isolated from the aerial parts with inflorescences of *Helichrysum italicum* in 2007, collected in Sardinia, Italy [1], this compound has attracted significant scientific interest due to its broad spectrum of biological activities, including anti-inflammatory [1,2], antioxidant [3,4], antibacterial [5,6,7], cytotoxic [4,8], neuroprotective [9], antiparasitic [8] activities, metabolic regulation [10], autophagy modulation [11], and cytoprotective effects [12,13]. While arzanol is commercially available from specialized suppliers (≥98% purity), its costly price reflects the present challenges of isolation from natural sources and synthesis. Despite the extensive documentation of arzanol’s diverse biological activities, some knowledge gaps remain. The substantial discrepancy between potent *in vitro* enzyme inhibition and the doses required for *in vivo* efficacy highlights pharmacokinetic limitations, particularly its extensive serum albumin binding that effectively sequesters the compound from target tissues. While arzanol’s conformational flexibility enables multitarget engagement, this complicates the determination of which mechanisms drive therapeutic effects in specific disease contexts. Key development requirements include derivatization strategies to maintain activity under physiological conditions while reducing protein binding, comprehensive structure-activity relationship studies to identify optimized analogs, metabolic stability profiling, and evaluation in chronic disease models. This review systematically examines the chemical characteristics, conformational behavior, biological activities, molecular targets, and pharmacokinetic properties of arzanol based on the published scientific literature from 2007 to 2025. A detailed compilation of biological activity assays with their corresponding experimental models and effective concentrations is tabulated in Table 1.

## 2. Literature Search Methodology

A comprehensive literature search was conducted using Google Scholar, PubMed, and Web of Science databases from 2007 (when arzanol was first characterized [1]) through 2025, regarding activities reported for the compound arzanol. The primary search term “arzanol” was used. Publications were included if they reported original experimental data on arzanol’s biological activities, described arzanol’s chemical characterization or synthesis, provided pharmacokinetic or mechanistic insights, were published in peer-reviewed journals, and were available in English. Studies were excluded if they only mentioned arzanol without experimental data, or were review articles without original data, although the latter were used for reference tracking and contextual information. All identified articles were manually assessed for relevance to the compound arzanol. Data were extracted systematically, including biological activity type, experimental model, effective concentrations such as IC_50_ and EC_50_ values, and mechanistic insights. Literature related to arzanol’s reported activities was also searched using the above-mentioned databases for the purpose of comparison and discussion.

## 3. Isolation, Natural Sources and Synthesis

Arzanol has been successfully isolated from two Mediterranean *Helichrysum* species: *H. italicum* (Roth) G. Don subsp. *microphyllum* [1,3,5,6] and *H. stoechas* (L.) Moench [15]. *H. italicum* and *H. stoechas* grow mainly in the Mediterranean basin, particularly abundant in countries such as Italy, France, Spain, Greece, and Croatia, where these species thrive naturally in dry, sandy and stony areas at a wide range of altitudes. Arzanol is predominantly found in the aerial parts and inflorescences of these plants, with yields varying significantly depending on the source material and extraction methodology. Initial isolation of arzanol was achieved by Appendino *et al.* (2007) [1] from *H. italicum* subsp. *microphyllum,* collected near Arzana (Sardinia, Italy) 0.097% *w*/*w* using acetone maceration. Subsequent studies have reported variable yields from the same subspecies: Rosa *et al.* (2007) obtained 0.081% *w*/*w* [3], while Taglialatela-Scafati *et al.* (2013) achieved a higher yield of 0.296% *w*/*w* [5]. Werner *et al.* (2019) reported a considerably lower yield of only 0.002% *w*/*w* [6]. *H. stoechas* has shown the highest arzanol content, with Les *et al.* (2017) isolating 0.48% *w*/*w* [15]. The extraction of arzanol from plant material typically involves maceration with acetone or methanol in cold or at room temperature, followed by various chromatographic purifications.

While arzanol is naturally occurring, its total synthesis has been achieved to support structure-activity relationship studies to provide a reliable supply for biological investigations. Minassi *et al.* (2012) [16] reported the first total synthesis of arzanol through a biomimetic approach that couples phloracetophenone with 6-ethyl-4-hydroxy-5-methyl-α-pyrone via a methylene bridge. Two synthetic strategies were developed, differing in the methylene source: the first employed paraformaldehyde as the methylene donor (61% yield), while the second utilized Eschenmoser’s salt (65% yield). The relatively straightforward synthesis, accomplished in moderate to good yields, makes arzanol and its derivatives accessible for extensive pharmacological evaluation without relying only on natural plant resources.

### Synthetic Derivatives

Several synthetic arzanol analogs have been developed. A series of arzanol derivatives was synthesized by condensing phloracetophenone with different pyrone cores and various aldehydes as linking agents, generating compounds with modified side chains and substitution patterns [16]. Notably, the hexyl-substituted analogs 6-diethyl-1′-hexyl-5,5,6-trimethylarzanol and 1′-hexylarzanol demonstrated enhanced mPGES-1 inhibitory potency (IC_50_ = 0.2–0.3 µM) compared to natural arzanol, while retaining good antibacterial activity against multidrug-resistant *Staphylococcus aureus*. However, these synthetic derivatives of arzanol have been relatively poorly explored regarding biological activity.

## 4. Chemical Structure and Properties

Arzanol represents a 3-prenylated acetophloroglucinol moiety linked to an α-pyrone unit through a methylene bridge. Despite its water insolubility, arzanol exhibits high solubility in polar organic solvents, including methanol, ethanol, dimethyl sulfoxide (DMSO), and acetone [16].

### Conformational Dynamics and Solution Structure

The solution structure of arzanol has been investigated through a combination of experimental NMR spectroscopy and computational density functional theory (DFT) calculations [17]. Arzanol possesses a dynamic structure resulting from multiple conformational processes. The phenolic hydroxyl groups adjacent to the methylene bridge in the phloroglucinol moiety form intramolecular hydrogen bonding with either the carbonyl or enolic hydroxyl group of the pyrone moiety, functioning alternately as hydrogen bond donors or acceptors. While arzanol theoretically has the potential to exist in either 2-pyrone or 4-pyrone configurations, detailed spectroscopic analysis confirmed that the 2-pyrone configuration represents the predominant roto-tautomeric state in solution [17]. This preference for the 2-pyrone form has important implications for the compound’s biological activity and molecular recognition (Figure 2).

The extensive hydrogen bond network within arzanol results in a notable reduction in overall polarity, a phenomenon analogous to that observed when comparing cyclic peptides to their linear counterparts [17]. This intramolecular hydrogen bonding effectively shields polar groups from the surrounding solvent, potentially influencing the compound’s membrane permeability and interaction with biological targets.

The conformational behavior of arzanol contrasts with that of the structurally related compound helipyrone, which exists in solution as essentially a single, monorotameric form. The presence of multiple geometric configurations enables these molecules to accommodate various binding sites on different macromolecular targets [17]. This conformational adaptability of arzanol is hypothesized to correlate with the diverse multitarget activities.

## 5. Anti-Inflammatory Activity

### 5.1. NF-κB Pathway Inhibition

Arzanol exhibits potent anti-inflammatory activity through multiple molecular targets and signaling pathways. Arzanol inhibits nuclear factor-κB (NF-κB) signaling pathway, with an IC_50_ of approximately 12 µM [1]. This inhibition of NF-κB, a master regulator of inflammatory responses, translates to broad suppression of downstream inflammatory mediators. Detailed dose–response studies demonstrated that arzanol effectively inhibits the production of key pro-inflammatory cytokines, including interleukin-1β (IL-1β), tumor necrosis factor-α (TNF-α), interleukin-6 (IL-6), and interleukin-8 (IL-8). Additionally, the compound suppresses the production of prostaglandin E_2_ (PGE_2_), a critical lipid mediator of inflammation. The IC_50_ values for these various inflammatory mediators range from 5.6 to 21.8 µM, indicating consistent and potent anti-inflammatory activity across multiple targets [1].

### 5.2. Dual Enzymatic Inhibition

At the molecular level, arzanol functions as a dual inhibitor of two key enzymes in the arachidonic acid cascade: microsomal prostaglandin E_2_ synthase-1 (mPGES-1) and 5-lipoxygenase (5-LOX). The compound inhibits mPGES-1 with remarkable potency (IC_50_ = 0.4 µM), making it one of the most potent natural mPGES-1 inhibitors identified to date. Its inhibitory potency places it in the same potency class as other prominent natural product inhibitors such as curcumin, garcinol, and boswellic acid derivatives, which typically report IC_50_ values in the sub-micromolar to low micromolar range [18,19,20,21]. Nevertheless, IC_50_ values across independent studies are prone to variations in assay protocols, enzyme sources, and glutathione concentration, which influence mPGES-1 inhibitor potency [18]. While synthetic inhibitors like MF63 [22] can demonstrate superior potency in isolated enzyme assays (IC_50_ in the low nanomolar range), its IC_50_ of 0.42 µM in A549 cells, is comparable to arzanol’s. Cellular permeability and other pharmacokinetic factors can limit the translation of exceptional enzyme potency into cellular efficacy. Importantly, mPGES-1 inhibition does not elevate asymmetric dimethylarginine (ADMA), a biomarker of cardiovascular toxicity linked to COX-2 inhibition, further supporting its safer pharmacological profile [23]. Unlike COX-2 inhibitors, which block all prostanoid production systemically, mPGES-1 inhibitors selectively target PGE_2_ synthesis while preserving other protective prostanoids. In mice, COX-2 inhibition with parecoxib significantly elevated plasma ADMA—an established predictor of cardiovascular risk—while mPGES-1 deletion did not affect ADMA levels [24]. Another study provides complementary evidence that mPGES-1 inhibitors maintain anti-inflammatory efficacy in preclinical models while potentially offering vascular protective effects [23]. Hence, mPGES-1 inhibition may represent a safer therapeutic strategy for treating inflammation compared to COX-2 inhibition, as it preserves the renal COX-2/ADMA regulatory axis while maintaining anti-inflammatory effects through selective PGE_2_ reduction. Similarly, arzanol inhibits 5-LOX with an IC_50_ of 3.1 µM, effectively blocking the production of pro-inflammatory leukotrienes [2,16]. This dual inhibition represents a distinct pharmacological profile compared to selective COX-2 or 5-LOX inhibitors. Svouraki *et al.* (2017) demonstrated that prenylated acetophenone dimers from *Acronychia pedunculata* exhibiting dual 5-LO/mPGES-1 inhibition showed superior anti-inflammatory efficacy with reduced side effects compared to single-target inhibitors [25].

### 5.3. In Vivo Validation

The anti-inflammatory efficacy of arzanol has been validated *in vivo* using a rat pleurisy model, a well-established model of acute inflammation. Administration of arzanol at 3.6 mg/kg significantly reduced inflammatory cell infiltration into the pleural cavity and decreased the levels of both PGE_2_ (47% reduction) and leukotrienes (31% reduction) in the inflammatory exudate [2]. These *in vivo* findings confirm that the dual inhibition of mPGES-1 and 5-LOX observed *in vitro* translates to meaningful anti-inflammatory effects in living organisms. The observation that arzanol demonstrates activity in both human cell-based assays and rodent models warrants further investigation, given documented species differences in mPGES-1 structure between humans and rodents [26]. Systematic comparison of arzanol’s binding affinity and inhibitory potency against purified human and rodent mPGES-1 would clarify whether this compound overcomes typical species selectivity barriers. The *in vivo* efficacy of arzanol (47% PGE_2_ reduction, 31% leukotriene reduction at 3.6 mg/kg) demonstrates superior potency compared to structurally related synthetic geranyl phloroglucinol analogs from *Acronychia pedunculata*, which required 10 mg/kg to achieve comparable anti-inflammatory effects in acute inflammation models [25].

## 6. Antioxidant Activity

### 6.1. Protection Against Lipid Peroxidation in Lipoproteins

The antioxidant properties of arzanol have been extensively characterized across multiple biological systems. Rosa *et al.* (2011) conducted comprehensive studies investigating the protective effects of arzanol against lipid peroxidation in biologically relevant systems [14]. In isolated human low-density lipoproteins (LDL), a critical target in atherosclerosis development, pretreatment with arzanol provided protection against copper-induced oxidative damage. The protection was dose-dependent, with significant effects observed starting at 8 µM concentration. Arzanol effectively preserved the levels of polyunsaturated fatty acids (PUFAs) and cholesterol while simultaneously inhibiting the formation of oxidative products, including hydroperoxides and oxysterols [14]. The oxidative degradation of unsaturated fatty acids and cholesterol in biological membranes and LDL contributes significantly to tissue damage and pathological processes, particularly in cardiovascular diseases and atherosclerosis. The experimental model using copper-induced LDL oxidation at physiological temperature (37 °C) mimics *in vivo* conditions, as atherosclerotic arterial walls contain trace copper ions capable of promoting LDL oxidation.

Comparative studies revealed that arzanol’s protective efficacy matched that of established antioxidant standards, including butylated hydroxytoluene (BHT), α-tocopherol, and the potent natural antioxidant curcumin. In liposome protection assays, arzanol demonstrated superior efficacy compared to other phenolic compounds, including prenyl curcumin, homogentisic acid, vanillin, and vanillyl alcohol [4,27,28,29,30].

### 6.2. Mechanistic Insights into Antioxidant Action

The antioxidant mechanism of arzanol involves multiple complementary pathways. The compound interacts with phospholipid polar heads, leading to accumulation at lipoprotein surfaces [14]. Direct radical scavenging occurs through hydrogen donation from the phenolic hydroxyl groups at the particle interface. Additionally, the multiple oxygenated functional groups in arzanol provide potential metal chelation sites. Computational DFT studies have elucidated the metal chelation mechanisms, identifying that the most stable arzanol-Cu^2+^ complex forms when the Cu^2+^ ion binds concurrently to the oxygen of the phenolic hydroxyl group neighboring the prenyl chain and to the π-bond of the prenyl chain itself [31]. A second highly favorable binding mode involves simultaneous coordination between the carbonyl oxygen of the α-pyrone ring and the oxygen of a nearby phenolic hydroxyl group.

### 6.3. Cellular Cytoprotection Against Oxidative Stress

Effective cellular cytoprotection occurs at 5–50 µM, with toxicity manifesting above 100 µM in normal cells, providing a good safety concentration margin. The protective concentrations scale proportionally with oxidative challenge—7.5 µM against 750 µM TBH [3] versus 50 µM against 2500–5000 µM H_2_O_2_ [12]. The antioxidant properties of arzanol extend beyond simple chemical systems to provide meaningful protection in cellular models. In VERO cells (monkey kidney fibroblasts), arzanol at 7.5 µM reduced tert-butyl hydroperoxide (TBH)-induced lipid peroxidation by 40%, with no cytotoxicity observed up to 40 µM [3]. At non-cytotoxic concentrations, arzanol exerted noteworthy protection against oxidative damage in both VERO cells and differentiated Caco-2 cells (human intestinal epithelial cells), significantly decreasing the formation of oxidative products [14].

#### 6.3.1. Protection of Skin Cells

Recent investigations have expanded the understanding of arzanol’s cytoprotective effects to dermatologically relevant models. Piras *et al.* (2024) investigated the protective effects against H_2_O_2_-induced oxidative stress in HaCaT human keratinocytes, a key model for skin cells [12]. Pre-incubation with arzanol (5–100 μM for 24 h) caused no cytotoxicity or morphological alterations. When keratinocytes were subsequently challenged with cytotoxic concentrations of H_2_O_2_ (2.5 and 5 mM), arzanol pretreatment provided significant protection against cell death. This protective effect was linked to arzanol’s ability to decrease intracellular reactive oxygen species (ROS) generation and inhibit lipid peroxidation under oxidative stress conditions. Arzanol pretreatment protected against H_2_O_2_-induced apoptosis, reducing the apoptotic cell population. This anti-apoptotic effect was directly associated with preservation of mitochondrial health, as arzanol prevented H_2_O_2_-induced mitochondrial membrane depolarization, maintaining membrane potential similar to healthy, unstressed cells [12].

#### 6.3.2. Neuroprotection Against Oxidative Stress

The cytoprotective effects of arzanol have also been demonstrated in neuronal models. Piras *et al.* (2024a) investigated protection in human neuroblastoma SH-SY5Y cells subjected to H_2_O_2_-induced oxidative stress [13]. In differentiated SH-SY5Y cells, which more closely resemble mature neurons, pretreatment with arzanol (5–25 μM) preserved both viability and morphology when exposed to cytotoxic H_2_O_2_ concentrations (0.25–0.5 mM), as assessed by MTT assay. While arzanol also conferred protection in undifferentiated SH-SY5Y cells, the efficacy was less pronounced. The compound further exerted anti-apoptotic effects, lowering the baseline apoptotic/necrotic cell fraction in untreated cultures and reducing H_2_O_2_-induced cell death, as measured by propidium iodide staining and caspase activation assays [13]. The differential protection between differentiated and undifferentiated SH-SY5Y cells (5–25 µM versus 5 µM only) [13] provides mechanistic insight into arzanol’s neuroprotective effects observed *in vivo* [9]. The preservation of mitochondrial membrane potential at antioxidant concentrations contrasts with the mitochondrial fragmentation observed at higher, cytotoxic doses [11], suggesting dose-dependent modulation of mitochondrial function. This response—protection at low concentrations, toxicity at high concentrations—aligns with arzanol’s selective cytotoxicity toward cancer cells, which operate with elevated baseline ROS and reduced antioxidant reserve capacity.

### 6.4. In Vivo Antioxidant Effects

The *in vivo* validation using iron-nitrilotriacetic acid (Fe-NTA) induced lipid peroxidation confirms that arzanol’s *in vitro* antioxidant activity translates to systemic protection. At 9 mg/kg, arzanol significantly reduced malondialdehyde and 7-ketocholesterol formation [4]. The less pronounced effect on hydroperoxide reduction suggests preferential protection against secondary oxidation products rather than primary radical formation. The antioxidant activity at these doses likely contributes to the anti-inflammatory effects observed in the rat pleurisy model at similar dosing (3.6 mg/kg) [2], as reduction in oxidative stress attenuates NF-κB activation [1].

## 7. Antimicrobial Activity

The antimicrobial properties of arzanol have been evaluated against various pathogenic microorganisms, with particular emphasis on antibacterial activity. Taglialatela-Scafati *et al.* (2013) reported potent antibacterial activity against a panel of six drug-resistant *Staphylococcus aureus* strains, with remarkably low minimum inhibitory concentration (MIC) values ranging from 1 to 4 µg/mL [5]. However, in a separate broad-spectrum antimicrobial screening, arzanol was found to be inactive at 20 µg/mL against *S. aureus* strain ATCC 25923, as well as against *Mycobacterium tuberculosis* and other tested pathogens [6]. This discrepancy in antibacterial activity may reflect several factors, including differences in assay conditions, culture media composition, or strain-specific susceptibility patterns. While the specific mechanism of arzanol’s antibacterial activity has not been elucidated, related phloroglucinol compounds act through membrane disruption, with some derivatives shown to bind phosphatidylglycerol and cause membrane depolarization [32,33]. Additional mechanisms reported for phloroglucinols include oxidative stress generation and effects on protein synthesis [34].

Other phloroglucinol α-pyrone derivatives isolated from *Helichrysum* species have shown limited antimicrobial potency in Gram-positive bacteria, and usually no activity against Gram-negative bacteria. For instance, helitalone B, isolated from the aerial parts of *H. italicum*, showed no activity at 20 µg/mL [6]. The arenol/homoarenol mixture, isolated from the aerial parts of *H. stoechas,* required 12–25 mg/L for weak activity against Gram-positive bacteria [35]. Helichrytalicine B, isolated from the aerial parts of *H. italicum,* exhibited bactericidal activity against *S. epidermidis* at 128 µg/mL [7]. Italipyrone (aerial part of *H. italicum*) showed MICs of 3–6 mg/L against Gram-positive bacteria with no Gram-negative activity [35]. However, acylphloroglucinol 3B, isolated from the flowers of *H. gymnocomum*, showed MIC 6.3 μg/mL against *S. aureus* ATCC 12600 and 7.8 μg/mL against MRSA ATCC 33592 [36]. A mixture of the dipyrones (two alpha pyrones conjugated through a methylene bridge), namely helipyrone A, B and C, achieved MIC values of 6 mg L^−1^ against *S. aureus* and *B. subtilis* and showed no activity against Gram-negative bacteria [35]. Helitalone A, another dipyrone, was found inactive at 20 µg/mL [6]. Crude extracts demonstrated broader spectrum activity than isolated compounds. Ethanolic extracts of *H. stoechas* achieved MIC values of 8 μg/mL against *S. aureus* ATCC 25923 and 32 μg/mL against *E. coli* ATCC 35218 and *P. aeruginosa* ATCC 10145 [37]. The antibacterial potential of hydroalcoholic extracts appears limited to affecting Gram-positive bacteria, with MIC values of 6–9 μg/mL for the most potent compounds, falling within clinically relevant ranges; moreover, the lack of biofilm activity and limited Gram-negative coverage restrict potential applications.

## 8. Molecular Targets and Mechanisms

### 8.1. Brain Glycogen Phosphorylase

Glycogen phosphorylase (GP) is the central enzyme responsible for glycogen mobilization (glycogenolysis) within cells. The brain isoform (bGP) is strongly and non-cooperatively activated by AMP and regulates glycogenolysis in accordance with local energy demands [38]. The glycogen that bGP mobilizes serves a dual function within tissues. On the one hand, it acts as a glucose reserve, supplying metabolites essential for energy production and the pentose phosphate pathway [39]. On the other hand, disruptions in glycogen metabolism leading to its excessive accumulation within cells are detrimental and contribute to the onset of glycogen storage diseases [40].

A study by del Gaudio *et al.* (2018) [10] utilized an MS-based proteomics to investigate the cellular interactome of arzanol and identify its direct molecular targets. This study identified brain glycogen phosphorylase (bGP), a key enzyme in glucose metabolism, as the main high-affinity target of arzanol. The identification was achieved using an experimental design employing arzanol-functionalized agarose beads to “fish” for binding partners from complex HeLa cell protein lysates. Arzanol protected bGP from proteolytic degradation in a dose-dependent manner, confirming specific binding.

Inhibition of glycogen mobilization in the brain has been linked to cognitive impairment and neurodegeneration [41]. In the brain, impaired glycogen mobilization has been associated with cognitive dysfunction and neurodegeneration [41,42,43]. Abnormal glycogen accumulation has been reported in various neurological and metabolic disorders, including amyotrophic lateral sclerosis, Alzheimer’s disease, diabetes, and aging [44,45,46]. Consequently, pharmacological stimulation of brain glycogen breakdown has been proposed as a potential therapeutic approach for neurological disorders [41].

Detailed characterization of the arzanol-bGP interaction [10] revealed that adenosine monophosphate (AMP), the natural allosteric activator of bGP, prevented arzanol from binding to the enzyme. This suggested that arzanol interacts with bGP at the same allosteric binding site as AMP. The study confirmed direct, high-affinity binding with a measured dissociation constant (Kd) of 0.32 ± 0.15 µM, indicating a physiologically relevant interaction [10]. These experimental findings were corroborated by computational molecular docking studies, which predicted that arzanol’s most favorable binding pose occurs within the AMP allosteric pocket. Conversely, only modest enzyme inhibition by arzanol was observed at high AMP concentrations [10].

Beyond binding characterization, arzanol increased the catalytic activity of bGP in a concentration-dependent manner in HeLa cell lysates. This establishes arzanol not as an inhibitor, but as a positive modulator (activator) of bGP, representing a novel activity for this class of phloroglucinol-pyrone compounds [10]. In particular, arzanol interacts with the residues Gln71, Asp227, Thr240 (by H-bonds) and Tyr75 (by an aromatic or π-interaction) [10].

Given that arzanol functions as a glycogen phosphorylase activator, it distinguishes mechanistically from inhibitors, more frequently reported in the natural products literature [47], like the flavonol quercetin [48], the polyphenol ellagic acid, anthocyanidins cyanidin and delphinidin [49], or anthraquinones [50]. The identification of arzanol as a positive modulator rather than inhibitor of bGP thus expands the known mechanistic diversity of natural products targeting glycogen phosphorylases. Given that activation of brain glycogen breakdown represents a potential therapeutic strategy to combat neurodegenerative conditions [41], arzanol provides an interesting scaffold for medicinal chemistry optimization and for validating bGP as a clinical target.

### 8.2. Autophagy Modulation and Mitochondrial Dysfunction

Deitersen *et al.* (2021) identified arzanol as a modulator of autophagy through flow cytometry-based screening of 300 natural compound library and subsequently characterized its anticancer effects and molecular mechanism of action [11]. This study revealed that arzanol exhibits complex, dual-stage effects on the autophagy pathway. Initially identified as an inhibitor of late-stage autophagic flux, this activity was confirmed by arzanol’s ability to cause significant accumulation of lipidated LC3-II and the autophagy receptor p62/SQSTM1 in HeLa cells. However, a more detailed investigation revealed that while inhibiting the final stages of autophagy, arzanol simultaneously acts as an inducer of early autophagosome biogenesis. This was evidenced by a significant increase in the number of ATG16L1-positive structures and the formation of numerous, albeit smaller-than-usual, autophagosomes [11]. While numerous phytochemicals have been documented to affect autophagy pathways in cancer cells [51,52,53,54], the simultaneous induction of early autophagosome biogenesis coupled with inhibition of late-stage autophagic flux positions arzanol within a select group of compounds exhibiting complex, stage-specific autophagy modulation. Most of the literature-described natural compounds that modulate autophagy are reported to be inducers and a few to be inhibitors [55]. Resveratrol is a well-established inducer of early autophagy, primarily through AMPK activation [56] and mTOR inhibition [57]. Curcumin [58] and schizandrin [59] are natural products reported to protect stress-induced excessive cell apoptosis by inhibiting AMPK/mTOR signaling-mediated autophagy. However, the combination of both activities within a single molecule is rarer and potentially more therapeutically interesting. It suggests a coordinated disruption that could be more effective in overwhelming cancer cell survival mechanisms dependent on functional autophagy, indicating that the context-dependent outcome—cytoprotective or cytotoxic—of autophagy modulation is a crucial consideration for therapeutic application.

The underlying mechanism for arzanol’s mitochondrial effects was identified as direct mitochondrial toxicity, characterizing it as a mitotoxin. Arzanol significantly reduced both basal and maximal cellular oxygen consumption rates (OCR) in HeLa cell cultures, suggesting a targeting of the oxidative phosphorylation (OXPHOS) system [11]. Specific targets were identified within the mitochondrial respiratory chain, including the Complex III proteins UQCRH and CYCS, and the Complex I-related protein NDUFS4 [11]. Furthermore, arzanol was found to target quinone-dependent reductases like NQO1 and other proteins such as CPS1, GARS, and HSPA9, highlighting its ability to disrupt multiple metabolic and survival pathways [11]. This broad targeting capacity is advantageous, as it may reduce the potential for cancer cells to develop single-target resistance.

The mitochondrial dysfunction induced by arzanol translates to meaningful anticancer activity. In urothelial bladder carcinoma models, arzanol exhibited moderate cytotoxicity as a single agent, with IC_50_ values of 6.6 µM in cisplatin-sensitive and 9.6 µM in cisplatin-resistant RT-112 cells [11]. Importantly, arzanol demonstrated significant activity as a chemosensitizer. When used in combination with the standard chemotherapeutic cisplatin, arzanol enhanced its potency approximately threefold, reducing cisplatin’s IC_50_ from 22.5 µM to 7.1 µM in sensitive cancer cells [11]. This ability to overcome chemoresistance aligns with the recognized strategy of combining standard agents with natural compounds to enhance efficacy and reverse resistance, as seen with agents like curcumin, which acts as a chemosensitizer by downregulating NF-B [60].

### 8.3. SIRT1 Inhibition and Metabolic Regulation

SIRT1 is the most studied of histone deacetylases (sirtuins) and acts as a key regulator of metabolic balance, inflammation, and cell survival. Both reduced and excessive SIRT1 activity can be detrimental. Given its role in major diseases such as diabetes, neurodegeneration, cardiovascular disorders, and cancer [61], identifying novel SIRT1 modulators remains a major therapeutic focus [62,63,64].

Borgonetti *et al.* (2023) identified sirtuin 1 (SIRT1) as another important molecular target of arzanol [9]. Molecular docking studies revealed that arzanol binds to SIRT1 through a network of four hydrogen bonds involving residues Arg274, Tyr280, Gln345, and Ile347. This predicted binding was validated through enzyme inhibition assays, which demonstrated that arzanol exhibits SIRT1 inhibitory activity comparable to or superior to nicotinamide (a known SIRT1 inhibitor) at concentrations of 10–100 µM. Moreover, in mouse hippocampal tissue, arzanol treatment reduced both SIRT1 protein expression and enzymatic activity, with concurrent downregulation of FOXO1 signaling, a key downstream target of SIRT1. This SIRT1 inhibition likely contributes to arzanol’s metabolic, neurobehavioral, and probably anticancer effects [9,65].

## 9. Pharmacokinetics and Bioavailability

### 9.1. Gastrointestinal Stability

A critical factor in the development of orally active natural products is their stability under gastrointestinal conditions. Silva *et al.* (2017) investigated the gastrointestinal stability of arzanol present in aqueous decoctions of *H. stoechas* using a validated *in vitro* digestion model [66]. The extracts containing arzanol were subjected to sequential treatment mimicking the digestive process: first with artificial gastric juice (pepsin buffered at pH 1.2) followed by artificial pancreatic juice (pancreatin buffered at pH 8), at physiological temperature (37 °C) for a total of 4 h. HPLC analysis comparing the extracts before and after the digestion process indicated stability. Arzanol was not enzymatically degraded or chemically modified during the simulated gastric and intestinal digestion. The chromatographic profile remained unchanged, and no loss of arzanol content was detected. These results provide evidence that following oral ingestion, arzanol is likely to reach the small intestine in its intact, bioactive form [66].

### 9.2. Intestinal Absorption and Cellular Transport

Rosa *et al.* (2011) provided insights into the bioavailability of arzanol using the Caco-2 cell monolayer model, the gold standard for predicting intestinal absorption of drug candidates [14]. The Caco-2 model mimics the human intestinal barrier, including the expression of relevant transporters and metabolic enzymes. Time-course studies revealed that arzanol accumulates inside Caco-2 epithelial cells in a time-dependent manner. After 4 h of incubation, approximately 34% of the applied arzanol was found within the cells, indicating significant cellular uptake. More importantly, transport studies demonstrated that arzanol was able to pass through the Caco-2 cell monolayer, providing direct evidence for its potential for oral absorption [14].

### 9.3. Serum Protein Binding and the Bi-Directional Implications for Activity and Delivery

The systemic efficacy of arzanol is directly impacted by its high affinity for serum proteins. In cell culture with serum-supplemented growth medium, arzanol was found to be ineffective and did not show biological activity (specifically, no autophagy modulation) [11]. Arzanol binds to serum proteins, especially albumin, which prevents it from entering the cells or reaching its intracellular targets. This property accounts for the lack of observable effects in earlier *in vitro* screenings performed in full (serum-containing) medium [11]. Conversely, under serum-free but amino acid-rich conditions, arzanol induced the accumulation of LC3-II—a hallmark of autophagy activation. This demonstrates that arzanol can trigger a starvation-like autophagic response, but only when its biological activity is not masked by the presence of serum proteins [11].

Albumin acts as a natural carrier protein in the blood of living organisms and extends the half-life of the albumin-bound drug and facilitates its accumulation in tumors or inflamed tissues via the enhanced permeability and retention (EPR) effect. Furthermore, this binding can help reduce toxicity and improve the stability of the drug [11]. Hence, a critical objective for pharmaceutical development is to derivatize arzanol in such a way that it keeps its serum-binding properties while regaining its biological activity [11].

### 9.4. Blood–Brain Barrier Penetration

While arzanol shows promising oral bioavailability, *in silico* pharmacokinetic modeling performed by Piras *et al.* (2024) [13] predicted poor blood–brain barrier (BBB) penetration. This limitation may restrict the compound’s direct effects on the central nervous system, although peripheral actions could still influence brain function through indirect mechanisms. The predicted poor BBB penetration contrasts with the observed neurobehavioral effects [13].

## 10. Neurobehavioral and Metabolic Effects Through SIRT1 Inhibition

Sirtuins are a class of histone deacetylase enzymes that control many fundamental cellular functions such as gene expression, metabolism, DNA repair, and mitochondrial function [64]. Sirtuin 1 (SIRT1) is the most extensively studied member of the sirtuin family. SIRT1 functions as a regulatory hub that helps cells maintain balance between metabolism, inflammation, and survival. Both insufficient and excessive SIRT1 activity can be harmful, so the direction of modulation that is beneficial depends on the state of the cell and tissue. Its regulatory influence on key chronic conditions, including diabetes, neurodegenerative, cardiovascular diseases, and tumorigenesis [61], makes SIRT1 a significant therapeutic target. Consequently, research efforts focus on identifying novel modulators of SIRT1 activity [62,63,64].

Arzanol was predicted *in silico* to bind to SIRT1 through a network of four hydrogen bonds involving residues Arg274, Tyr280, Gln345, and Ile347 [9]. Moreover, enzyme inhibition assays demonstrated that arzanol exhibits SIRT1 inhibitory activity comparable to or superior to nicotinamide (a known SIRT1 inhibitor) at concentrations of 10–100 µM. *Ex vivo* studies using mouse hippocampal tissue confirmed the physiological relevance of SIRT1 inhibition. Arzanol treatment reduced both SIRT1 protein expression and enzymatic activity, with concurrent downregulation of FOXO1 signaling, a key downstream target of SIRT1 [9].

SIRT1 activity in hippocampal neurons mediated anxiety and depressive-like behaviors in mice [67]. Overexpression of SIRT1 has been linked to anxiety, and depression [68] and brain-specific *Sirt1*-knockout mice showed reduced susceptibility to depression [69]. Borgonetti *et al.* (2023) conducted *in vivo* studies investigating the neurobehavioral effects of a methanolic extract of *H. stoechas* inflorescences, with arzanol as the principal bioactive constituent [9]. Repeated oral administration of the extract (100 mg/kg daily for 3 weeks) in mice produced significant anxiolytic effects comparable to the benzodiazepine diazepam and antidepressant-like effects comparable to the tricyclic antidepressant amitriptyline. Importantly, these beneficial effects were achieved without the common side effects associated with conventional anxiolytics and antidepressants. The arzanol-containing extract did not impair locomotor activity or memory function, suggesting a favorable safety profile for potential therapeutic use [9].

Moreover, the mice fed with *H. stoechas* inflorescence (100 mg/kg for 3 weeks) significantly reduced their body weight gain without reducing food intake [9]. This suggests that arzanol may influence metabolic processes, potentially through its effects on SIRT1 and related metabolic pathways. The ability to prevent weight gain without affecting appetite could potentially have important implications for metabolic health and obesity management.

In addition to systemic neurobehavioral effects, arzanol demonstrated direct neuroprotective properties *in vitro*. Exposure of SH-SY5Y cells to glutamate lead to 1.5-fold overexpression of SIRT1 in the control group. This effect was reduced by pretreatment with *H. stoechas* extract with doses from 1 to 100 μg/mL reducing SIRT1 protein content in a dose-dependent manner. In search for the main constituent(s) that might be responsible for this activity, molecular docking studies suggested arzanol as the most promising candidate for SIRT1 inhibition, which was then validated through enzyme inhibition assays [9].

At concentrations of 5–10 μM, arzanol protected SH-SY5Y neuroblastoma cells against glutamate-induced excitotoxicity [9]. Glutamate excitotoxicity is a major mechanism of neuronal death in various neurological conditions including stroke, traumatic brain injury, and neurodegenerative diseases. The ability of arzanol to protect against this form of neuronal damage suggests potential applications in neuroprotection.

## 11. Cytotoxicity Profile and Selectivity

Arzanol’s role as a chemosensitizer fits within the broader context of using bioactive compounds from medicinal plants for drug discovery [70]. The administration of multiple chemotherapeutic drugs with different targets, known as combined chemotherapy, is established practice for improving efficacy and reducing adverse effects [60]. Experimental evidence suggests that combining antitumoral agents with natural compounds, such as curcumin [71], resveratrol [72], and epigallocatechin-3-gallate [73], has the potential to reduce cancer treatment resistance and perform chemoprotective actions. This beneficial effect is generated by different mechanisms, including increased tumoricidal effect via sensitization of cancer cells, reversing chemoresistance through inhibition of drug resistance targets, and decreasing chemotherapy-induced toxicity in non-tumoral cells by promoting repair mechanisms [60,70].

The cytotoxic effects of arzanol have been evaluated across multiple cancer cell lines to assess its potential as an anticancer agent and to establish its selectivity profile. Rosa *et al.* (2017) conducted detailed dose–response studies revealing selective cytotoxicity [4]. Arzanol exhibited dose-dependent cytotoxicity against HeLa cells (human cervical carcinoma) and B16F10 cells (murine melanoma). However, the most pronounced effects were observed in Caco-2 colon cancer cells. Importantly, no significant cytotoxicity was observed at concentrations below 100 μM in normal cell lines, indicating a favorable therapeutic window [4].

The SIRT1 inhibition observed for arzanol by Borgonetti *et al.* (2023) [9] may represent a key mechanistic contributor to its anticancer effects. Lin *et al.* (2022) [65] demonstrated that both heliomycin (H1) and its derivative HDR2 bind to and inhibit SIRT1 while exhibiting marked antiproliferative activity against cancer cells. Given that SIRT1 post-translationally modifies numerous substrate proteins across diverse cellular pathways, precise control of its activity could serve as an effective therapeutic strategy in cancer treatment.

Arzanol, as H1 and HDR2, modulates SIRT1 and autophagy pathways [9,11,65]. Arzanol induces early autophagosome biogenesis while blocking late-stage autophagic flux [11]. SIRT1 downregulation by H1 [65] appears to occur via autophagic degradation and/or proteasomal degradation. Arzanol’s reduction in both SIRT1 protein expression and enzymatic activity, with concurrent downregulation of FOXO1 signaling, provides a plausible link between SIRT1 inhibition and the selective cytotoxicity observed by Rosa *et al.* (2017) [4], though direct validation in cancer cell models remains necessary to confirm this mechanistic connection.

However, SIRT1’s biological role varies substantially between different tissues and cellular environments [74]. In neural tissue, SIRT1 is often neuroprotective, and its inhibition affects cognition, metabolism, and stress responses [75]. In cancer, SIRT1 can be either tumor-promoting or tumor-suppressing, depending on the specific cancer type and stage [75]. Hence, while arzanol inhibits SIRT1 in hippocampal tissue [9] with resultant effects on FOXO1 signaling, the consequences of this same molecular interaction in cancer cells may differ. FOXO1 is a critical regulator of apoptosis, cell cycle progression, and DNA repair. SIRT1 deacetylates and activates FOXO1, so SIRT1 inhibition would be expected to reduce FOXO1 activity [9]. In cancer cells, modulation of SIRT1-FOXO1 could affect tumor cell survival and proliferation [76]. Arzanol’s dual effect on autophagy—simultaneously inducing early autophagosome biogenesis while blocking late-stage flux—might be partially mediated through SIRT1- FOXO1, though this remains speculative without direct experimental evidence [11]. Nevertheless, whether arzanol’s SIRT1 inhibition contributes to its anticancer effects requires cancer cell-specific studies examining SIRT1 and FOXO1 expression and activity following arzanol treatment, complemented measuring SIRT1 overexpression or comparison with selective SIRT1 inhibitors. Given the multi-target nature of most phytochemicals and the stage-specific autophagy modulation exhibited by arzanol [11], SIRT1 inhibition may be a contributor to arzanol’s overall anticancer profile.

## 12. Prospects for Drug Development

Arzanol faces several challenges that must be addressed for successful drug development. One obstacle is its high serum protein binding, particularly to albumin. Under serum-containing conditions, arzanol loses its biological activity as the protein binding prevents cellular uptake and target engagement. While albumin binding can benefit drug delivery, it limits bioavailability. As noted, a development objective is to derivatize arzanol to retain beneficial serum-binding properties while regaining biological activity. The pharmacokinetic challenges extend beyond protein binding. Despite demonstrated gastrointestinal stability and intestinal absorption in Caco-2 models supporting oral bioavailability, there exists a huge gap between potent enzyme inhibition in isolated systems and the doses required for *in vivo* anti-inflammatory effects. This highlights ADME issues that require optimization. The poor water solubility, while addressable through formulation strategies, adds complexity to pharmaceutical development. The predicted poor blood–brain barrier penetration, if true, may limit direct central nervous system applications despite observed neurobehavioral effects. Another challenge involves translating preclinical findings to human therapeutics. There are species differences in mPGES-1 structure between humans and rodents. While arzanol demonstrates activity in both human cell-based assays and rodent models, systematic comparison of binding affinity and inhibitory potency against purified human versus rodent mPGES-1 would be necessary to confirm that this compound overcomes typical species selectivity barriers that have hindered other mPGES-1 inhibitor development programs. The multitarget nature of arzanol, while therapeutically advantageous for efficacy, complicates regulatory development pathways. Demonstrating which of the multiple mechanisms—NF-κB inhibition, dual mPGES-1/5-LOX inhibition, glycogen phosphorylase activation, SIRT1 inhibition, or autophagy modulation—predominates in specific therapeutic contexts requires extensive mechanistic studies in disease-relevant models. The tissue- and context-dependent effects of SIRT1 modulation exemplify this complexity: SIRT1 inhibition may be beneficial in hippocampal tissue for anxiety and depression but could have different consequences in cancer cells or cardiovascular tissue. Manufacturing and supply represent practical considerations. While the synthesis developed by Minassi *et al.*, 2012 [16] provides reliable access to arzanol in moderate yields (61–65%), scaling to industrial production levels for clinical trials would require process optimization. The reliance on natural plant sources (yields of 0.002–0.48% *w*/*w* from *Helichrysum* species) is insufficient for drug development, making synthetic production essential.

## 13. Conclusions

Arzanol represents a remarkable example of a multitarget natural product with diversly characterized biological activities spanning anti-inflammatory, antioxidant, neuroprotective, and anticancer activities. Its conformational flexibility arising from intramolecular hydrogen bonding [17] enables interaction with diverse molecular targets including mPGES-1, 5-LOX, NF-κB, brain glycogen phosphorylase, and SIRT1. The dual inhibition of mPGES-1 and 5-LOX [2,16] represents a therapeutically advantageous profile, as the partial rather than complete mPGES-1 inhibition may preserve physiological prostanoid functions while avoiding the cardiovascular toxicity associated with COX-2 inhibitors [18,23,24]. *In vivo* validation in rat models confirmed meaningful anti-inflammatory efficacy, producing significant reductions in both PGE_2_ and leukotrienes [2].

The mechanistic complexity of arzanol extends beyond simple enzyme inhibition. Its identification as a brain glycogen phosphorylase activator (Kd = 0.32 µM) [10] positions it uniquely among natural products, which more commonly inhibit this enzyme, and suggests potential therapeutic applications in neurological disorders characterized by glycogen accumulation [41]. Arzanol’s SIRT1 inhibitory activity [9] provides a mechanistic framework linking its diverse biological effects, from the anxiolytic and antidepressant activities observed in behavioral studies [9] to its potential anticancer mechanisms. Arzanol’s ability to simultaneously induce early autophagosome formation while blocking late-stage autophagic flux [11] represents an unusual mechanistic profile that distinguishes it from most natural autophagy modulators. This complex autophagy modulation, combined with direct mitochondrial toxicity and demonstrated chemosensitizing activity against cisplatin-resistant cells [11], suggests therapeutic potential beyond single-target agents.

Combined with its favorable safety profile, including selectivity for cancer cells over normal cells at therapeutic concentrations [4], demonstrated gastrointestinal stability [66], and intestinal absorption in Caco-2 models [14], arzanol’s multitarget mechanism positions it as a promising candidate for complex disorders. However, arzanol faces the classic challenge of natural product development: the thousand-fold gap between its potent enzyme inhibition in isolated systems and the doses required for anti-inflammatory effects *in vivo* [2,4] highlights critical obstacles related to absorption, distribution, metabolism, and particularly its high serum protein binding [11]. Despite poor water solubility [16], oral delivery remains feasible through appropriate formulation strategies. Future structure-activity relationship studies, metabolic profiling, tissue distribution analyses, and evaluation in chronic disease models will be essential to advance arzanol from a biochemically interesting scaffold toward clinical therapeutic development.

## Figures and Tables

**Figure 1 plants-14-03474-f001:**
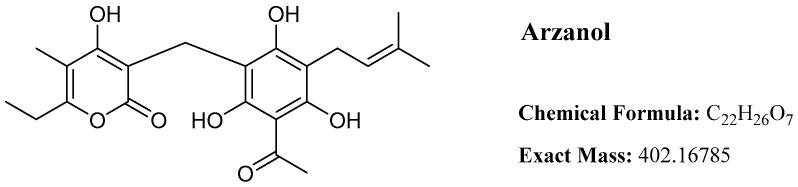
Structural formula of arzanol.

**Figure 2 plants-14-03474-f002:**
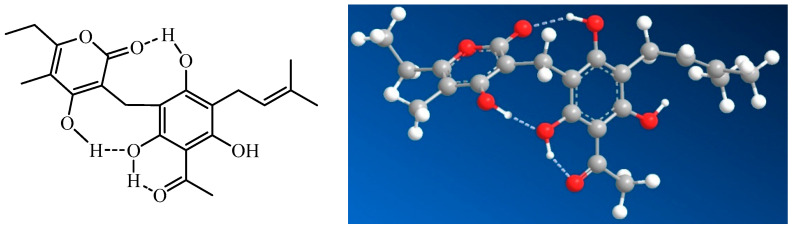
Intramolecular hydrogen bonding in arzanol, according to the study of Rastrelli *et al*. (2016) [17].

**Table 1 plants-14-03474-t001:** Summary of arzanol’s biological activities, effective concentrations, and experimental models.

Category	Subcategory	Biological Activity	IC_50_/EC_50_/Dose	Model/Cell Type	Ref.
Anti-inflammatory	NF-κB Pathway	Switches off NF-κB in Jurkat immune-cell assay	IC_50_ ~5 µg/mL (≈12 µM)	Jurkat cells	[1]
Anti-inflammatory	Cytokine Inhibition	IL-1β reduction	IC_50_ 5.6 µM	LPS-stimulated human monocytes	[1]
Anti-inflammatory	Cytokine Inhibition	TNF-α reduction	IC_50_ 9.2 µM	LPS-stimulated human monocytes	[1]
Anti-inflammatory	Cytokine Inhibition	IL-6 reduction	IC_50_ 13.3 µM	LPS-stimulated human monocytes	[1]
Anti-inflammatory	Cytokine Inhibition	IL-8 reduction	IC_50_ 21.8 µM	LPS-stimulated human monocytes	[1]
Anti-inflammatory	Eicosanoid Pathway	PGE_2_ reduction	IC_50_ 18.7 µM	LPS-stimulated human monocytes	[1]
Anti-inflammatory	Eicosanoid Pathway	5-LOX inhibition	IC_50_ 3.1 µM	Recombinant enzyme assay	[2]
Anti-inflammatory	Eicosanoid Pathway	5-LOX inhibition in neutrophils (A23187/AA stimulation)	IC_50_ 2.9 µM	Human neutrophils	[2]
Anti-inflammatory	Eicosanoid Pathway	5-LOX inhibition in neutrophils (LPS/fMLP stimulation)	IC_50_ 8.1 µM	Human neutrophils	[2]
Anti-inflammatory	Eicosanoid Pathway	mPGES-1 inhibition	IC_50_ 0.4 µM	IL-1β-stimulated A549 cells	[2]
Anti-inflammatory	Eicosanoid Pathway	COX-1 inhibition (12-HHT reduction)	IC_50_ 2.3 µM	Human platelets	[2]
Anti-inflammatory	Eicosanoid Pathway	COX-1 inhibition (TXB_2_ reduction)	IC_50_ 2.9 µM	Human platelets	[2]
Anti-inflammatory	Eicosanoid Pathway	PGE_2_ reduction in whole blood	~50% at 30 µM	Human whole blood	[2]
Anti-inflammatory	*In vivo*	Reduces pleural fluid volume	59% at 3.6 mg/kg	Rat pleurisy model	[2]
Anti-inflammatory	*In vivo*	Reduces inflammatory cell infiltration	48% at 3.6 mg/kg	Rat pleurisy model	[2]
Anti-inflammatory	*In vivo*	Reduces PGE_2_ levels	47% at 3.6 mg/kg	Rat pleurisy model	[2]
Anti-inflammatory	*In vivo*	Reduces LTB_4_ levels	31% at 3.6 mg/kg	Rat pleurisy model	[2]
Antioxidant/Cytoprotective	Lipid Protection	Complete inhibition of linoleic acid autoxidation	IA50 1.2 nmol	Solvent-free film (37 °C, 32 h)	[3]
Antioxidant/Cytoprotective	Lipid Protection	Protection under EDTA-mediated oxidation	≥80% at 1 nmol, 100% at ≥2.5 nmol	Linoleic acid + EDTA	[3]
Antioxidant/Cytoprotective	Lipid Protection	Protection in FeCl_3_-catalyzed oxidation	13% at 40 nmol, 80% at 80 nmol	Linoleic acid + FeCl_3_	[3]
Antioxidant/Cytoprotective	Cholesterol Protection	Complete protection against cholesterol oxidation	100% at 10 nmol	140 °C thermal oxidation	[3]
Antioxidant/Cytoprotective	Cholesterol Protection	Prevention of 7-keto and 7β-OH formation	IA50 5.6–6.8 nmol	140 °C, 1–2 h	[4]
Antioxidant/Cytoprotective	LDL Protection	Protects LDL from Cu^2+^-induced oxidation	Significant from 8 µM	Human LDL (37 °C, 2 h)	[14]
Antioxidant/Cytoprotective	Liposome Protection	Protection of polyunsaturated fatty acids	89.1% at 10 µM	Phospholipid liposomes + Cu^2+^	[4]
Antioxidant/Cytoprotective	Cell Protection	No cytotoxicity to VERO cells	Up to 40 µM	VERO fibroblasts	[3]
Antioxidant/Cytoprotective	Cell Protection	Reduces lipid peroxidation in VERO cells	40% reduction at 7.5 µM	VERO cells + 750 µM TBH	[3]
Antioxidant/Cytoprotective	Cell Protection	No cytotoxicity to VERO cells	Up to 50 µM	VERO cells (24 h)	[14]
Antioxidant/Cytoprotective	Cell Protection	No cytotoxicity to differentiated Caco-2	Up to 100 µM	Differentiated Caco-2 (24 h)	[14]
Antioxidant/Cytoprotective	Cell Protection	Protects VERO cells from TBH	Significant at 25–50 µM	VERO cells + TBH	[14]
Antioxidant/Cytoprotective	Cell Protection	Protects Caco-2 cells from TBH	Significant from 25 µM	Differentiated Caco-2 + TBH	[14]
Antioxidant/Cytoprotective	*In vivo*	Prevents plasma lipid consumption	9 mg/kg *i.p.*	Wistar rats + Fe-NTA	[4]
Antioxidant/Cytoprotective	*In vivo*	Protects plasma unsaturated fatty acids	Complete at 9 mg/kg	Wistar rats + Fe-NTA	[4]
Antioxidant/Cytoprotective	*In vivo*	Reduces plasma MDA levels	9 mg/kg	Wistar rats + Fe-NTA	[4]
Antioxidant/Cytoprotective	*In vivo*	Reduces plasma 7-ketocholesterol	9 mg/kg	Wistar rats + Fe-NTA	[4]
Antioxidant/Cytoprotective	Keratinocyte Protection	No cytotoxicity	5–100 µM	HaCaT cells	[12]
Antioxidant/Cytoprotective	Keratinocyte Protection	Protects against H_2_O_2_ cytotoxicity	50 µM pre-treatment	HaCaT cells + 2.5–5 mM H_2_O_2_	[12]
Antioxidant/Cytoprotective	Keratinocyte Protection	Reduces ROS generation	50 µM	HaCaT cells + 0.5–5 mM H_2_O_2_	[12]
Antioxidant/Cytoprotective	Keratinocyte Protection	Prevents lipid peroxidation	50 µM	HaCaT cells + 2.5–5 mM H_2_O_2_	[12]
Antioxidant/Cytoprotective	Keratinocyte Protection	Prevents apoptosis (caspase-3/7)	50 µM	HaCaT cells + 5 mM H_2_O_2_	[12]
Antioxidant/Cytoprotective	Keratinocyte Protection	Preserves mitochondrial membrane potential	50 µM	HaCaT cells + 5 mM H_2_O_2_	[12]
Antioxidant/Cytoprotective	Neuronal Protection	Increases viability (differentiated cells)	5–25 µM	Differentiated SH-SY5Y	[13]
Antioxidant/Cytoprotective	Neuronal Protection	Cytotoxic at high dose (differentiated cells)	71% reduction at 100 µM	Differentiated SH-SY5Y	[13]
Antioxidant/Cytoprotective	Neuronal Protection	Non-toxic (undifferentiated cells)	2.5–100 µM	Undifferentiated SH-SY5Y	[13]
Antioxidant/Cytoprotective	Neuronal Protection	Protects differentiated cells from H_2_O_2_	5–25 µM	Differentiated SH-SY5Y + 0.5 mM H_2_O_2_	[13]
Antioxidant/Cytoprotective	Neuronal Protection	Protects undifferentiated cells from H_2_O_2_	5 µM	Undifferentiated SH-SY5Y + 0.5 mM H_2_O_2_	[13]
Antioxidant/Cytoprotective	Neuronal Protection	Reduces ROS (differentiated cells)	5–25 µM	Differentiated SH-SY5Y + H_2_O_2_	[13]
Antioxidant/Cytoprotective	Neuronal Protection	Reduces ROS (undifferentiated cells)	5 µM	Undifferentiated SH-SY5Y + H_2_O_2_	[13]
Antioxidant/Cytoprotective	Neuronal Protection	Decreases basal apoptosis	5–25 µM	Differentiated SH-SY5Y	[13]
Antioxidant/Cytoprotective	Neuronal Protection	Protects from H_2_O_2_-induced apoptosis (PI assay)	10–25 µM	ifferentiated SH-SY5Y + 0.25 mM H_2_O_2_	[13]
Antioxidant/Cytoprotective	Neuronal Protection	Protects from H_2_O_2_-induced apoptosis (caspase)	10–25 µM	Differentiated SH-SY5Y + 0.25 mM H_2_O_2_	[13]
Neuroprotective/Neurobehavioral	Neuroprotection	Protects from glutamate toxicity	5–10 µM	SH-SY5Y neuroblastoma cells	[9]
Neuroprotective/Neurobehavioral	Metabolic Effects	Attenuates weight gain (no effect on food intake)	100 mg/kg, 3 weeks oral	Mice	[9]
Neuroprotective/Neurobehavioral	Behavioral Effects	Anxiolytic effect (comparable to diazepam)	100 mg/kg	Mice	[9]
Neuroprotective/Neurobehavioral	Behavioral Effects	Antidepressant effect (comparable to amitriptyline)	100 mg/kg	Mice	[9]
Neuroprotective/Neurobehavioral	Behavioral Effects	No locomotor or memory impairment	100 mg/kg	Mice	[9]
Neuroprotective/Neurobehavioral	SIRT1 Modulation	Binds SIRT1 via 4 H-bonds	Molecular docking	*in silico*	[9]
Neuroprotective/Neurobehavioral	SIRT1 Modulation	SIRT1 inhibition	10–100 µM	Cell-free assay	[9]
Neuroprotective/Neurobehavioral	SIRT1 Modulation	Reduces SIRT1 expression/activity	100 mg/kg HSE or 10 µM	Mouse hippocampus (*ex vivo*)	[9]
Neuroprotective/Neurobehavioral	SIRT1 Modulation	Suppresses FOXO1 signaling	100 mg/kg HSE or 10 µM	Cell and tissue studies	[9]
Anticancer	Autophagy Modulation	Inhibits starvation-induced autophagy	+164.6% mCitrine-LC3	Mouse embryonic fibroblasts	[11]
Anticancer	Autophagy Modulation	Accumulates LC3-II and p62/SQSTM1	Not specified	HeLa cells	[11]
Anticancer	Autophagy Modulation	Increases the number but reduces the size of autophagosomes	Not specified	HeLa cells	[11]
Anticancer	Autophagy Modulation	Increases ATG16L1-positive structures	Not specified	HeLa cells	[11]
Anticancer	Direct Cytotoxicity	Cytotoxic to cisplatin-sensitive bladder cancer	IC_50_ 6.6 µM	RT-112 cells	[11]
Anticancer	Direct Cytotoxicity	Cytotoxic to cisplatin-resistant bladder cancer	IC_50_ 9.6 µM	RT-112 cells	[11]
Anticancer	Direct Cytotoxicity	Cytotoxic to undifferentiated Caco-2	55% reduction at 100 µM	Undifferentiated Caco-2	[4]
Anticancer	Direct Cytotoxicity	Cytotoxic to HeLa cells	36% reduction at 200 µg/mL	HeLa cells (24 h)	[4]
Anticancer	Direct Cytotoxicity	Cytotoxic to B16F10 cells	95% reduction at 200 µg/mL	B16F10 melanoma (24 h)	[4]
Anticancer	Direct Cytotoxicity	No cytotoxicity to differentiated Caco-2	Up to 100 µM	Differentiated Caco-2	[4]
Anticancer	Direct Cytotoxicity	No cytotoxicity to L5178Y cells	20 µg/mL	Murine lymphoma L5178Y	[6]
Anticancer	Chemosensitization	Enhances cisplatin effect	Reduces IC_50_ from 22.5 to 7.1 µM	RT-112 cells	[11]
Anticancer	Mitochondrial Effects	Induces mitochondrial fragmentation	Not specified	HeLa cells	[11]
Anticancer	Mitochondrial Effects	Reduces basal/maximal respiration	Not specified	HeLa cells	[11]
Anticancer	Mitochondrial Effects	Inhibits OXPHOS complexes II, III, V	Not specified	Isolated mitochondria	[11]
Anticancer	Mitochondrial Effects	Inhibits NQO1	Significant at 10 µM	Direct activity assay	[11]
Metabolic	Glycogen Metabolism	Binds brain Glycogen Phosphorylase (bGP)	Identified as main target	HeLa cell lysates	[10]
Metabolic	Glycogen Metabolism	Direct bGP binding (DARTS confirmed)	Dose-dependent protection	DARTS assay	[10]
Metabolic	Glycogen Metabolism	Binds at AMP allosteric site	Competitive with AMP	Competitive binding assay	[10]
Metabolic	Glycogen Metabolism	High-affinity bGP binding	Kd = 0.32 ± 0.15 µM	Surface Plasmon Resonance	[10]
Metabolic	Glycogen Metabolism	Predicted AMP site binding	Kd,pred = 0.65 ± 0.18 µM	Molecular docking	[10]
Metabolic	Glycogen Metabolism	Activates bGP enzyme	Dose-dependent activation	HeLa cell lysates	[10]
Antibacterial	*S. aureus* (drug-resistant)	SA1199B (NorA efflux) inhibition	MIC 1 µg/mL	*S. aureus* SA1199B	[5]
Antibacterial	*S. aureus* (drug-resistant)	XU212 (TetK) inhibition	MIC 4 µg/mL	*S. aureus* XU212	[5]
Antibacterial	*S. aureus* (drug-resistant)	ATCC 25923 (reference) inhibition	MIC 1 µg/mL	*S. aureus* ATCC 25923, Mueller-Hinton/MTT method	[5]
Antibacterial	*S. aureus* (drug-resistant)	RN4220 (MsrA) inhibition	MIC 2 µg/mL	*S. aureus* RN4220	[5]
Antibacterial	*S. aureus* (drug-resistant)	EMRSA-15 (epidemic MRSA) inhibition	MIC 4 µg/mL	*S. aureus* EMRSA-15	[5]
Antibacterial	*S. aureus* (drug-resistant)	EMRSA-16 (epidemic MRSA) inhibition	MIC 2 µg/mL	*S. aureus* EMRSA-16	[5]
Antibacterial	Negative result	No activity against *S. aureus* ATCC 25923	No inhibition at 20 µg/mL	*S. aureus* ATCC 25923, CLSI broth microdilution	[6]

## Data Availability

No new data were created or analyzed in this study. Data sharing is not applicable to this article.

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
