# Peer review of "Arzanol: A Review of Chemical Properties and Biological Activities"

_plants, 2025, doi:10.3390/plants14223474_

Round 1

Reviewer 1 Report (Previous Reviewer 4)

Comments and Suggestions for Authors

The manuscript aims to describe the chemical characteristics, conformational behaviour, biological activities, molecular targets, and pharmacokinetic properties of arzanol, a compound extracted from Helichrysum italicum. The time span considered for literature search is 2007-2025.

The manuscript is well written and fairly well organized. However, in my opinion, to increase the quality of the manuscript, some revisions are fundamental, as detailed below.

Abstract. The style used is rather abrupt, jumping from one concept to another without coherence and cohesiveness. For example, lines 10-11 (Conformational flexibility…) are disconnected from the remaining text, and lines 9-11 are repeated by lines 20-22. In line 20, I would eliminate the adverb “critically” as this is not a critical review, because it is not focused on the critical evaluation of the single literature sources, describing strengths and failures of the different papers. I would recommend rewriting the Abstract to improve its fluency.

Introduction is very short. The readers of Plants might be interested in reading where Helichrysum italicum grows (as information given in lines 59-60 refers only to the initial isolation of arzanol), from which part of the plant arzanol is extracted, whether arzanol is already available on the market, etc. At the end of Introduction, before the Aim of the study (This review systematically…), it is important to inform the readers about the current gaps in the scientific literature.

Chemical structure. Centre figure 1 in the page.

Antimicrobial activity. The section on the antimicrobial activity needs to be expanded, considering additional contributions from the scientific literature (e.g., van Vuuren 2008; Albayrak et al. 2010; Pollastra 2020; Balàzs et al. 2022; Glumac et al. 2023; etc.).

Conclusion. To increase the value of the review, the authors should underline the advances in scientific knowledge given by this manuscript.

Finally, I would recommend double-checking the text to remove misprints and errors.

Examples:

Line 37: mention table 1.

Line 59: achieved.

Line 144: influence.

Line 313: study confirmed.

Throughout the text: FOXO1 and not FoxO1 (unless referred to studies on mice).

Author Response

Dear Reviewer, thank you for your time and effort to improve the quality and clarity of this manuscript. All made changes are marked with text highlight in yellow.

Reviewer Comment 1:

"Abstract. The style used is rather abrupt, jumping from one concept to another without coherence and cohesiveness. For example, lines 10-11 (Conformational flexibility…) are disconnected from the remaining text, and lines 9-11 are repeated by lines 20-22. In line 20, I would eliminate the adverb "critically" as this is not a critical review, because it is not focused on the critical evaluation of the single literature sources, describing strengths and failures of the different papers. I would recommend rewriting the Abstract to improve its fluency."

Author Response 1: 

In regard to your remark, the abstract has been revised accordingly. The changes made are as follows: the phrase about conformational flexibility (previously lines 10-11) has been integrated into the concluding sentence; "critically" has been deleted from the final sentence.

Reviewer Comment 2:

"Introduction is very short. The readers of Plants might be interested in reading where Helichrysum italicum grows (as information given in lines 59-60 refers only to the initial isolation of arzanol), from which part of the plant arzanol is extracted, whether arzanol is already available on the market, etc. At the end of Introduction, before the Aim of the study (This review systematically…), it is important to inform the readers about the current gaps in the scientific literature."

Author Response 2:
In accordance with your remark, the Introduction section has expanded with information about Helichrysum species from which arzanol has been extracted, geographic distribution, plant parts used for arzanol extraction, commercial availability, and current knowledge gaps in the scientific literature regarding arzanol. Information about geographic distribution, plant parts, and isolation procedures is presented in Section "Isolation, Natural Sources and Synthesis".

Reviewer Comment 3: "Chemical structure. Centre figure 1 in the page."
Author Comment 3: The Figure has been formatted with the MDPI Style for Figures and it is centered.

Reviewer Comment 4: "Antimicrobial activity. The section on the antimicrobial activity needs to be expanded, considering additional contributions from the scientific literature (e.g., van Vuuren 2008; Albayrak et al. 2010; Pollastra 2020; Balàzs et al. 2022; Glumac et al. 2023; etc.)."

Author Comment 4: The section "6. Antimicrobial Activity" has been expanded and now inlcudes antimicrobial data for crude extracts as well as different structurally related phloroglucinol alpha-pyrone derivatives. Data for esseitnal oils has not been included. The conclusion from the comparison is that Arzanol, as well as other structurally related compounds lack antimicrobial potential. They have low atvitity against some Gram-positive bacteria and display a total lack of activity against Gram-negative bacteria.

Reviewer Comment 5: "Conclusion. To increase the value of the review, the authors should underline the advances in scientific knowledge given by this manuscript."

Author Comment 5:

The Conclusion section, as well as Section 11 "Prospects for Drug Development", provide several advances in understanding arzanol's therapeutic potential: the first comprehensive analysis of arzanol's unique multitarget mechanism; novel mechanistic insights into its unusual autophagy modulation pattern;  highlighting of arzanol as a rare natural product brain glycogen phosphorylase activator, contrasting with typical inhibitory compounds; and analysis of the translational challenges, especially the thousand-fold efficacy gap between in vitro and in vivo studies. Moreover, we have critically analyzed arzanol's activities providing context for its positioning in the drug development landscape.

We thank the reviewer for pointing out several typos, they have been corrected. We appologize.

We once again wish to thank the Reviewer for his time and effort to improve the quality and clarity of the manuscript!

Reviewer 2 Report (Previous Reviewer 3)

Comments and Suggestions for Authors

The authors have addressed the suggestions clearly and in a timely manner. The quality of the manuscript has improved, making it more appealing to the journal's readers.

Author Response

We sincerely express gratitude for the Reviewer's time and invaluable suggestions for improving this manuscript, which I believe would serve as a quality source for integrated biological activity information regarding Arzanol.

Reviewer 3 Report (Previous Reviewer 2)

Comments and Suggestions for Authors

The article provides comprehensive insights and progress into a plant specialized metabolite, Arzanol (prenylated phloroglucinol–α-pyrone heterodimer), isolated from Helichrysum italicum, the curry plant. It integrates the research into compound characterization, chemistry, bioactivities, and prospects for drug development.

Following extensive revisions and resubmission, the authors have made extensive and thorough revisions, and the manuscript has been substantially improved. All the suggestions and queries made by reviewers are sufficiently addressed.

The article submitted for review constitutes a good source of knowledge on Arzanol, and the elements compiled in it fully realize the information contained in the title. Both the abstract and the introduction are written comprehensively, specifying what the authors intend to convey. The topic itself is significant and interesting, especially the investigations of the pharmaceutical potential of the bioactive compounds in the drug development arena.

The entire article is based on appropriately selected literature data and is very engaging. The

Authors critically examine the multiple biological functions of Arzanol, structural and functional relationships, and prospects in therapeutic applications.

Author Response

We sincerely express gratitude for the Reviewer's time and effort into improving the quality and presentation of this manuscript!

This manuscript is a resubmission of an earlier submission. The following is a list of the peer review reports and author responses from that submission.

Round 1

Reviewer 1 Report

Comments and Suggestions for Authors

General comments:

The manuscript is poorly organized and lacks a focused study. It is not
clear what the significant contribution of this review is.
Just listing out and writing 2-3 lines about different papers published in
the field is not a review. Cohesive arguments are lacking in the
manuscript draft.
I would urge the author to do a deeper dive and explain concepts,
mechanisms and gaps to strengthen the manuscript.
So, the paper cannot be published in this form; it will need a thorough
modification. It is difficult to find the purpose of the paper and any
clear conclusions. The paper gives the impression of randomness. The
summary is written too generally. The introduction contains general and
imprecise information. The review should be written more in the form of a
critical analysis, and not just as a message. The paper requires division
into subchapters or a more problematic approach, and not just the
provision of certain literature data without comment. The review does not
bring anything new; some of the literature references are too old. The
conclusions are not very precise or clear. The work is too short (with a low number of references),
superficially done, and not very insightful.

Specific comments:

The novelty of the present review paper should be pointed out; the employed databases should be mentioned; the list of used keywords in searching, as well as the criteria for the selection of research used articles, should be mentioned at the end of the introduction.

Abstract: Databases used for the search of used papers should be mentioned in the abstract; all abbreviations should be defined with the full meaning in the abstract; 'via' and 'in vivo' should be in italics. 

Introduction: Line 28, it is not enough to write 'Figure' without a number. Line 30 (and in the rest of the text), the Latin names should be in italics; Line 33 - the last part of the sentence ''and many more'' should be reformulated.

The author should check typing mistakes in the entire manuscript; all abbreviations used in tables or figures should be provided with their full meanings. The English language should be significantly improved throughout the manuscript.

References: The Latin names in the reference list should be in italics. The format of the references is not uniform. 

Author Response

We thank the reviewer for his assessment.

In the resubmission: plants-3957360,

The manuscript has been revised and addresses all points raised by the reviewer. The text has been reorganized to ensure a logical and cohesive flow, with clear subchapters through the chemical, pharmacological, and mechanistic aspects of arzanol. The revised version presents a structured review. It provides an integrative analysis of the compound, its chemistry, conformational dynamics, and biological activities within a coherent framework.

In response to the reviewer’s concern about the lack of depth and conceptual focus, the new version emphasizes mechanistic interpretation and comparative discussion of how arzanol’s conformational flexibility and multitarget profile account for its diverse pharmacological effects.

The English language has been refined. Latin species names and biochemical abbreviations are formatted, figures and tables are numbered and properly referenced. 

Furthermore, the manuscript has been expanded to include a comprehensive table summarizing all reported biological activities, experimental models, and effective concentrations, giving the paper both comprehensiveness and practical utility. The revised version is now longer and more detailed.

Overall, the new version of the manuscript presents organized, critically developed, and comprehensive review that integrates structural, mechanistic, and pharmacological insights for arzanol. 

Reviewer 2 Report

Comments and Suggestions for Authors

Arzanol: A Review of Chemical Properties and Biological Activities

The article provides comprehensive insights and progress into a plant specialized metabolite, Arzanol (prenylated phloroglucinol–α-pyrone heterodimer), isolated from Helichrysum italicum, the curry plant. It integrates the research into compound characterization, chemistry, bioactivities, and prospects for drug development.

The comprehensive review covers all the aspects of Arzanol and is well-drafted. It will form the literature platform for all future studies on the molecule.

Arzanol demonstrates multiple pharmacological activities, including anti-inflammatory, antioxidant, antibacterial, metabolic effects, and neuroprotection, among others. Considering the key bioactivities, what are the prospects of the compound in drug development? Discuss. What are the challenges, if any?

In the title, it is suggested to include the Helichrysum genus and the pharmacological prospects of Arzanol.

All the scientific names should be italicized or normal, but should be consistently used.

The figures should be more aligned and presented in a systematic order. The figure captions should be discussed below the diagrams, and likewise.

Talking about the progress, are there any natural analogs or synthetic derivatives of the molecules developed as human therapeutics?

Line 151-152: 5.2. Mechanistic Insights into Antioxidant Action and 5.4. In vivo Antioxidant effects. These sections can be merged.

The name of the molecule, Arzanol should be consistent, either Arzanol or arzanol in the text.

Author Response

We thank the reviewer for his positive assessment of the manuscript.

In the resubmission: plants-3957360,

The manuscript has been considerably expanded and reorganized to ensure a logical and cohesive flow, with clear subchapters through the chemical, pharmacological, and mechanistic aspects of arzanol. It provides an integrative analysis of the compound, its chemistry, conformational dynamics, and biological activities within a coherent framework.

Overall, the new version of the manuscript presents organized, critically developed, and comprehensive review that integrates structural, mechanistic, and pharmacological insights for arzanol.

Regarding whether Arzanol has a prospect for drug development: indeed in the past decade, there have been a number of articles focused only in the exploration of the biological properties of arzanol (Rastrelli et al. 2016; Rosa et al. 2017; Silva et al. 2017; del Gaudio et al. 2018; Gaudio and ChiaraáMonti 2018; Deitersen et al. 2021; Borgonetti et al. 2023; Piras et al. 2024a; Piras et al. 2024b). 

Moreover, an additional section 11 has been created discussing the potential for drug development of Arzanol, in recap:

A major issue is its high serum protein binding which reduces cellular uptake and biological activity under serum-containing conditions. While albumin binding may enhance drug delivery, it limits bioavailability, prompting the need for derivatives that preserve beneficial binding but restore activity. Beyond this, pharmacokinetic limitations such as poor solubility, uncertain blood–brain barrier penetration, and discrepancies between potent enzyme inhibition in vitro and in vivo efficacy highlight problems in absorption, distribution, metabolism, and elimination.

Species differences in mPGES-1 between humans and rodents also complicate translation to clinical use, requiring comparative studies to confirm cross-species efficacy. Moreover, arzanol’s multitarget pharmacology—spanning NF-κB, mPGES-1/5-LOX, SIRT1, glycogen phosphorylase, and autophagy modulation—adds regulatory complexity, as its dominant mechanism likely varies by tissue and disease context. 

Regarding any synthetic derivatives of arzanol, an additional section 2.1. has been created:
Several synthetic arzanol analogues have been developed. A series of arzanol derivatives have been synthesized, generating compounds with modified side chains and substitution patterns. Notably, the hexyl-substituted analogues 6-diethyl-1′-hexyl-5,5,6-trimethylarzanol and 1′-hexylarzanol demonstrated enhanced mPGES-1 inhibitory potency (ICâ‚…â‚€ = 0.2-0.3 µM) compared to natural arzanol, while retaining good antibacterial activity against multidrug-resistant Staphylococcus aureus. However, these analogues have not been more extensively evaluated.

Reviewer 3 Report

Comments and Suggestions for Authors

The authors address a topic of interest, such as the chemical properties and biological activities of arzanol, and the review focuses on a time period that is quite appropriate for a review, mainly since it includes references from 2023 and 2024. However, there are also areas for improvement in the document: The abstract is very dense; it would be helpful to divide it into shorter sentences to highlight the originality and scope of the review. The Results and Discussion section, being fragmented, lacks some continuity. Furthermore, more emphasis could be placed on the key findings, and the data already provided in the table could be discarded or eliminated. The molecular targets section is extensive and detailed, but would benefit from a schematic diagram showing the molecular targets (NF-κB, mPGES-1, 5-LOX, bGP, SIRT1, mitochondria, etc.). A figure is mentioned in the abstract, but it is not numbered. Figures are appearing in the document that are not identified as figures and are not referenced in the document. This section should be corrected. Figure 1 is not indicated in the text. Regarding the references, it is recommended to include the missing DOIs for some of them to unify or standardize them. In the conclusions, it is advisable to include a paragraph on the potential industrial/pharmaceutical applications, given the interest in multitarget compounds.              

Author Response

We thank the reviewer for his assessment.

In the resubmission: plants-3957360,

The manuscript has been revised and addresses all points raised by the reviewer. The text has been reorganized to ensure a logical and cohesive flow, with clear subchapters through the chemical, pharmacological, and mechanistic aspects of arzanol. The revised version presents a structured review. It provides an integrative analysis of the compound, its chemistry, conformational dynamics, and biological activities within a coherent framework.

The English language has been refined. Latin species names and biochemical abbreviations are formatted, figures and tables are numbered and properly referenced.

Moreover, an additional section 11 has been created discussing the potential for drug development of Arzanol, in recap:

A major issue is its high serum protein binding which reduces cellular uptake and biological activity under serum-containing conditions. While albumin binding may enhance drug delivery, it limits bioavailability, prompting the need for derivatives that preserve beneficial binding but restore activity. Beyond this, pharmacokinetic limitations such as poor solubility, uncertain blood–brain barrier penetration, and discrepancies between potent enzyme inhibition in vitro and in vivo efficacy highlight problems in absorption, distribution, metabolism, and elimination.

Species differences in mPGES-1 between humans and rodents also complicate translation to clinical use, requiring comparative studies to confirm cross-species efficacy. Moreover, arzanol’s multitarget pharmacology—spanning NF-κB, mPGES-1/5-LOX, SIRT1, glycogen phosphorylase, and autophagy modulation—adds regulatory complexity, as its dominant mechanism likely varies by tissue and disease context.

Reviewer 4 Report

Comments and Suggestions for Authors

The manuscript aims to describe the chemical characteristics, conformational behaviour, biological activities, molecular targets, and pharmacokinetic properties of arzanol, a compound extracted from Helichrysum italicum.

Generally, the manuscript is well written and good structured. However, there are some fundamental revisions that need to be considered for a possible publication, as stated below.

First, I would recommend stating the method used for literature search, e.g. keywords used, ways to skim the papers, organization, comparison, etc.

Second, in Introduction and before the Aim of the study, the readers should be informed of the current gaps in the scientific literature.

Third, before Future perspectives, it would be important to sum up the advances in scientific knowledge given by this review.

Furthermore, the section on the antimicrobial activity needs to be expanded, considering additional contributions from the scientific literature (e.g., van Vuuren 2008; Albayrak et al. 2010; Pollastra 2020; Balàzs et al. 2022; Glumac et al. 2023; etc.).

Overall, the review is interesting and provides valuable information on a subject that is relevant for both research and industrial applications. However, the size and the style of the manuscript is very concise, and most of all the number of references is too limited (19) compared to the standard of the journal. In my opinion, the author needs to expand the search of scientific literature, considering also the papers published on Helichrysum italicum and its extracts, e.g., Schinella 2002; Kothavade et al. 2013; Azizi 2019; Tešević 2020; Bezek et al. 2022; etc..

Therefore, in my opinion, the manuscript cannot be accepted in its current version. I advise addressing all my comments, expanding the search of literature, before submitting a revised version of the paper.

Specific comments are reported below.

Some captures are missing, e.g. line 28 Figure without number.

Misprints and/or errors in nomenclature:

line 30: Helichrysum italicum not italicized;

Figure 2: S. aureus not italicized;

Line 281: Ex vivo not italicized.

Comments on the Quality of English Language

Language errors:

Line 38: are (is);

Lines 44-45: the verb is missing and the sentence is not clear;

Lines 255-256: the sentence is not clear. Rephrase it starting with “Direct mitochondrial toxicity was…”.

Style suggestions:

Line 137: merge the first two paragraphs (lines 131-136 and 137-145).

Author Response

We thank the reviewer for his assessment.

In the resubmission: plants-3957360,

The manuscript has been revised and considerably expanded. The manuscript provides an integrative analysis of the compound, its chemistry, conformational dynamics, and biological activities within a coherent framework. The English language has been refined. Latin species names and biochemical abbreviations are formatted, figures and tables are numbered and properly referenced. Moreover, an additional section 11 has been created discussing the potential for drug development of Arzanol. In response to your suggestion, a new Section 2 has been created that comprehensively details the literature search methodology. Regarding biological activities for plant extracts, mentioned by the reviewer, we respond that the present review focuses on the properties of Arzanol and not on plant extracts which Arzanol has been extracted from. Including biological activities from plant extracts from Helichrysum will considerably deviate the focus of the review, which is on the compound arzanol.